# Analysis of the Psychosocial Impact of the COVID-19 Pandemic on the Nursing Staff of the Intensive Care Units (ICU) in Spain

**DOI:** 10.3390/healthcare10050796

**Published:** 2022-04-25

**Authors:** María Muñoz-Muñoz, Jesús Carretero-Bravo, Celia Pérez-Muñoz, Mercedes Díaz-Rodríguez

**Affiliations:** 1Vall d’Hebrón Hospital, 08035 Barcelona, Spain; muniozmaria@gmail.com; 2Department of Nursing and Physiotherapy, University of Cádiz, 11009 Cádiz, Spain; jesus.carretero@uca.es (J.C.-B.); celia.perez@uca.es (C.P.-M.)

**Keywords:** COVID-19, pandemic, coronavirus, ICU, nurses, psychosocial impact

## Abstract

(1) The public health emergency, caused by COVID-19, has resulted in strong physical and mental exhaustion in healthcare workers. This research has been designed with the aim to describe the psychosocial impact of the COVID-19 pandemic on nurses working in intensive care units (ICU) and identify the related risk factors. (2) This is a cross-sectional study, in which a self-administered questionnaire was designed to cover the dimensions of interest associated with psychosocial factors during the pandemic and their factor risks. (3) A total of 456 nursing professionals participated, and 88.4% were women. Most of the professionals had a temporary contract (71.3%) and person at risk close to them (88.8%). Regarding psychosocial factors, there was a worsening in most of the associated variables, especially in sleep problems, anxiety, stress, and job performance. Female nurses were more prone to anxiety. Those under 30, as well as those with temporary contracts, were more unfocused. Professionals with a person at risk in their environment felt much more worried. The degree of exposure was associated with greater fear. (4) Those nurses who were female, younger, and with a temporary employment contract were shown to be more vulnerable to the impact of the pandemic on their psychosocial health. Because of this, it is necessary to adopt effective strategies for the protection of nurses’ health, focusing on the specific risk factors identified.

## 1. Introduction

The health situation caused by the new SARS-CoV-2 coronavirus (hereafter COVID-19) has caused a drastic change in many aspects of our daily and working lives. Since 14 March 2020, the day on which the state of alert was declared in our country, Spain [1], the pandemic has brought about an unprecedented change that dramatically affected the Spanish healthcare system.

The first three waves in Spain took place between March 2020 and March 2021. By that date, the analysis reported to RENAVE identified 2,944,732 cases of COVID-19 in Spain. Of the cases, 7.2% were hospitalised, 0.7% were admitted to the ICU, and 1.4% died [2].

In Spain, during this period, the number of healthcare professionals infected with COVID-19 was one of the highest in the world. According to the 75th report on the COVID-19 situation in Spain, on 21 April 2021, 79,524 healthcare professionals were diagnosed with COVID-19 after 10 May 2020 [2]. In total, since the start of the pandemic, 169,694 cases of infected healthcare workers have been detected [3]. This is compounded by the inequalities and shortcomings of the Spanish health system [4], which add to the pandemic situation and impact on health professionals.

Another aspect observed in our country has been the stigmatisation and aggression towards healthcare workers by society, which puts added pressure on them and results in an aggravation of mental disorders [5]. The pandemic has also caused moral damage to professionals, as resources have been overstretched, due to the high number of patients; this has forced professionals to make difficult decisions, such as choosing which patients would not receive life support [6].

In a recent research, Lázaro-Pérez et al. [7] asked themselves whether anxiety had been generated in healthcare workers, concerning the processes of death in the pandemic. Significantly high levels of anxiety were found, with variables associated with this anxiety, such as the scarcity of personal protective equipment and lack of knowledge about COVID-19, i.e., variables that were more present in the Spanish healthcare system [4].

Likewise, a study (also carried out in Spain) at the beginning of the pandemic [8] analysed post-traumatic stress, anxiety, and depression among healthcare workers. Of those surveyed, 56.6% showed symptoms of post-traumatic stress disorder, 58.6% anxiety disorder, 46% depressive disorder, and 41.1% felt emotionally exhausted.

In the research carried out by Alonso et al. [9], which studied the prevalence of mental disorders and related factors in healthcare professionals in Spain in the first wave, with more than 9000 participants, it was found that half of the respondents suffered from a mental disorder and one in seven healthcare professionals presented a probable disabling mental disorder.

Similarly, the narrative review by Giorgi et al. [10] provides evidence that this pandemic has led to sleep disorders and suicidal thoughts in health workers, as well as high levels of psychological distress, insomnia, alcohol, and drug abuse, among others.

This same evidence is also expressed in the review by Villca et al. [11], indicating that it is frontline health professionals who suffer the most from symptoms of psychological distress, such as insomnia, anxiety, fear of contagion, post-traumatic stress disorder, suicidal ideation, and depression. Additionally, not only frontline professionals were affected, but also those professionals with very close contact with their patients’ airways, such as dentists, who have shown increased symptoms of psychological distress during the pandemic [12]. This shows the need to specifically analyse all health fronts associated with this pandemic.

More specifically, if we talk about nursing staff, Serrano-Ripoll et al. [13] analysed the effect of the COVID-19 pandemic on the psychological distress of nursing staff, confirming that stressors such as work overload, insufficient preparation, lack of support, deaths, and fear of infection had a positive and significant relationship with psychological distress. In the same vein, Del Pozo-Herce et al. [14] reported a prevalence of mental health risk factors in nursing professionals of 90%, with a predominance of feelings of burnout and emotional overload.

As has been seen, several studies have shown that the prevalence of stress, anxiety, depression, and social problems among frontline healthcare workers is high [15,16]. However, few studies have focused on analysing, more specifically, the impact on the psychosocial health of healthcare professionals in intensive care units in Spain, as well as understanding psychosocial health as the health associated with the emotional, mental, and social dimensions of the individual health. This is especially important in nurses, who have been essential in the hospital care of the most seriously ill patients and for whom psychosocial health protection measures must be established as a priority.

Therefore, this research has been designed with the aim to describe the impact of the COVID-19 pandemic on the psychosocial health of the nurses working in intensive care units (ICU), identify the related risk factors to detect the harm suffered by these workers, and be able to establish preventive measures to minimise this impact.

## 2. Materials and Methods

### 2.1. Aims and Study Design

The study’s aim was to analyze the psychosocial impact of the COVID-19 pandemic on ICU nurses in Spain. This information would help design protocols and risk control procedures, promoting the acquisition of coping strategies.

We performed an observational, descriptive, and cross-sectional study. The study was conducted between February and April 2021.

### 2.2. Population and Scope of the Study

The study was carried out throughout hospitals in Spain. The study’s target population was nurses in ICU that had worked in this position during some of the first three waves of the COVID-19 pandemic in Spain.

The following inclusion criteria were derived from these considerations: being nursing personnel in an ICU and having worked for at least one of the first three waves of the COVID-19 pandemic in such a unit in a hospital in Spain.

Participants who did not express their consent to participate in the study at the beginning of the questionnaire were excluded.

### 2.3. Instrument, Data Collection, and Procedure

A self-administered questionnaire was designed to carry out the research. For the development of the instrument, a previous literature review was carried out that identified the main variables that have affected the psychosocial health of healthcare workers during the pandemic [17,18], as well as their associated risk factors [19,20]. Following this review, the questionnaire was developed by a team, consisting of a medical researcher, psychologist, nurse, and mathematician.

It was preferred to construct a questionnaire from scratch, rather than using already validated questionnaires (because of the special characteristics of the pandemic), and focus the questions on those aspects that were considered most important, in the opinion of the research team. Taking the nursing staff’s knowledge of health issues into account, simple and direct questions on psychosocial health facts were chosen, in order to shorten the response time of the questionnaire. Certain terms, such as anxiety, fear, and stress, were clearly explained in the question. The complete contents are attached as Appendix A.

The first section of the instrument presents questions on the socio-demographic aspects (age, sex, affiliation, and period worked). This was followed by questions associated with the dimensions of interest in psychosocial health, mainly associated in this pandemic with working conditions, mental health, and change in social relations. These questions can be summarised as follows:Perception of working conditions during the pandemic: five questions (four yes/no questions and one with five options) aimed to determine the nurses’ opinions regarding the management of the pandemic in their unit, in terms of material and personnel support.Work relationships: three questions (two yes/no and one with different answer options) about the impact on the relationship between co-workers, difficulty in concentrating, and impact on work performance and, therefore, the health care provided.Perception of the state of mental health: we inquired regarding the feelings and mental states experienced by the health workers, such as worry, mental fatigue, sleep problems, fear, anxiety, psychological stress, and depression. In addition, nurses were also asked about psychological or psychiatric help (if they had to seek it, as well as if they were aware of the free services offered by the hospital) and the use of psychotropic medication. This section consisted of 11 questions, of which, eight had different options, and three were yes/no questions.Life events: whether they have suffered any traumatic event related to the pandemic, such as personal or family infection, admission, or death, due to COVID-19. It consisted of two yes/no response items.Social relations: focused on determining the social impact that the situation has had on professionals and their immediate environment. It was about the fear of infecting someone close to them, as well as whether drastic measures have been used to avoid it, they had people in their environment who were at risk of the disease, they were at risk, or they felt their relationships have been affected. This section consists of five yes/no response items.

The questionnaire was distributed online. Google Forms platform was used, under the protection of an academic email account of the university. To disseminate the questionnaire, a mailing was sent to the institutional accounts of healthcare professionals, and it was disseminated through the organisations and associations related to these professionals. Social networks were used as a means of publicising the research. * *

### 2.4. Study Variables

Primary variables in the study were associated with the psychosocial dimensions: health workers’ perception of working conditions during the pandemic, perception of the state of mental health, life events, and work relationships. There were qualitative variables with several options (two to five options). Most of them are worded to order the categories from highest to lowest.

Secondary variables were socio-demographic and particular characteristics of the pandemic, which allowed us to analyse the characteristics of the professionals who have completed the questionnaire. The socio-demographic variables were one continuous variable, i.e., age, as well as five qualitative variables: gender (male, female), link with the health care provider system (temporary, permanent, interim), period worked (first wave, second wave, third wave, all), person at risk nearby (no, yes), and degree of exposition (very low, low, medium, high, very high).

### 2.5. Statistical Procedures

Firstly, a descriptive analysis of the data obtained for the study variables was carried out. Those secondary variables with categories with a frequency of less than 5% were grouped, with age in three categories (under 30, between 30 and 40, over 40) and degree of exposition in three categories (low to medium, high, very high).

Within the primary variables, 12 questions were selected that, in the opinion of the research team, presented results of interest. The items were tested for their suitability for factor analysis using the KMO statistic (suitable for values greater than 0.7) and Bartlett’s test of sphericity [21]. After this, factor analysis was carried out, taking into account the criterion of eigenvalues greater than one for the selection of factors and performing a Quartimax rotation on the components. The reliability of the items was also analysed through the internal consistency of the responses. Cronbach’s alpha coefficient was calculated, where values above 0.7 are acceptable [22].

Each of the 12 questions was dichotomised into two categories and associated with the secondary variables through contingency tables. To see if there was an association between the responses and socio-demographic variables, the chi-squared test was performed, using Fisher’s test correction, when the conditions of the contingency table were not suitable for the chi-squared test. The means of the dimensions derived from the factor analysis were also compared, according to the secondary variables. The appropriate test was used, according to the situation (*t*-test with two means and ANOVA with more than two means).

This analysis was carried out with the SPSS statistical software (version 24). In all statistical tests, the standard level of 0.05 was taken as the significance level.

### 2.6. Ethical Considerations

The questionnaire was anonymous and did not collect personal data that could identify the participant. In addition, to preserve the confidentiality of the data, the IP address of the computer was deleted immediately. The information was treated confidentially and anonymously, since they had dissociated data, following the Data Protection Regulation (EU) 2016/679 of the European Parliament and Spanish Organic Law 3/2018.

In the survey presentation, participants were informed that completion of the questionnaire was entirely voluntary. Informed consent to participate in the study was collected to allow answer the questionnaire.

## 3. Results

The characteristics of the sample are shown in Table 1. In total, 456 nursing professionals completed the survey. Most participants were women (88.4%), with a low percentage of men (11.6%). Three subgroups were classified, according to age, with the majority being under 30 years of age (62.1%).

As for the link with the health care provider system, 71.3% had a temporary contract, in line with those provided during the pandemic. A total of 15.4% were interim, and 13.4% had a permanent position. On the other hand, 41.9% of those surveyed worked in all pandemic waves, while 58.1% did not. As for COVID-19 infections, 22.1% of the professionals were infected at the time of the survey.

Looking at the variables associated with the workplace, the degree of exposure was perceived as very high by 63.6% of respondents. In comparison, it was only low or very low for 3.1% of respondents. On the other hand, the relationship with colleagues was perceived as “better” by 77.2%, the same as always by 13.6%, and worse by 9.2%. As for the working conditions, 78.7% thought they were not suitable for coping with the situation, compared to 21.3% who thought they were. Almost all the nurses, 98.5%, also thought that the health system was not prepared for the situation. Finally, 37.1% considered that they had experienced a traumatic event related to the pandemic.

The second section of the survey was the workers’ psychosocial health state perception. The results of this section are shown in Table 2. A total of 78.5% felt much more worried for various reasons, such as for their own health, family’s health, work issues, or any aspect related to the pandemic. Only 2% said they were equally or less worried.

A total of 53.1% of the respondents stated that they had experienced difficulties focusing at work. In addition, 63.8% thought that their work performance has been negatively affected, impacting patient care. A total of 81.8% reported feeling much more tired, 89.9% had sleep problems, and 53.5% felt much more anxious than usual. In addition, 82.5% of the nurses who took part in the survey reported feeling down, depressed, or hopeless. Despite this, 53.1% denied having felt that they had lost their vocation, even though 52.2% considered that they had felt much more psychological stress than they usually do.

A total of 56.8% have felt much more afraid than usual, while the fear of infecting someone in their close family or social environment was experienced by 97.6% of respondents. A total of 92.8% of respondents felt that their relationships had been affected by the pandemic, and 94.1% had to take isolation measures because of their immediate environment. Additionally, 47.6% were unaware of the services offered to professionals in their hospital. Finally, 18.2% have increased their consumption of psychotropic drugs, and 7.2% use the same as always.

The psychometric properties of the primary variables of the questionnaire were tested. The KMO statistic and Bartlett’s test of sphericity showed a good fit for the factor analysis (KMO = 0.782 and *p*-value = 0.000, respectively). The factor analysis performed captured three dimensions, containing 59.2% of the variance explained. The association of the dimensions to the questions can be seen in Table 3. The first dimension is associated with fear and worry, the second with psychological aspects, and the third with variables associated purely with work. Cronbach’s alpha coefficient was 0.749, indicating adequate internal consistency.

Chi-square association tests were performed to check the association between nurses’ personal variables and dichotomized questions about psychosocial outcomes due to the pandemic. The results of the tests can be seen in Table 4, and the percentages of variation of the psychological variables between the different groups in Figure 1 and Figure 2.

The gender of the nurses influenced anxiety. Female nurses were more prone to anxiety, 59.3%, compared to 43.4% for males (*p*-value = 0.028).

The age variable was more associated with work-related variables. Those under 30 were more unfocused (57.2% vs. 33.3%, *p*-value = 0.005), noted a greater negative impact, and showed a greater loss of vocation (49.5% vs. 27.8%, *p*-value = 0.011). Younger people, with 85.2%, were also more depressed than those over 40 years of age, with 64.8% (*p*-value = 0.001).

Of interest is the employment relationship, which affected three variables, two of them closely related to the job itself. Those on temporary contracts felt less tired (*p*-value = 0.006) but were, nevertheless, more unfocused on their work: 56% of those on temporary contracts, compared to only 36.1% of those on permanent contracts (*p*-value = 0.016).

Those who worked during all waves of the pandemic felt much more tired than usual (77.0%), although this percentage was significantly higher for those who did not work during all waves (85.3%, *p*-value = 0.023).

Regarding professionals who had a person at risk in their environment, they felt much more worried than those who did not (80.2% vs. 64.7%, *p*-value = 0.011). They also represented a high percentage (*p*-value = 0.035) of the need for psychological help (“much more” (43.7%) and “somewhat more” (42.7%)). However, the negative impact on work was significantly higher (*p*-value = 0.021) in those who did not have a person at risk in their environment (78.4%) than those who did (62.0%).

The degree of exposure was associated with two variables. A total of 56.3% of nurses who considered their degree of exposure “very high” suffered psychological stress, compared to 34.7% of those who considered their exposure the same or lower (*p*-value = 0.015). The degree of exposure was associated with greater fear: while those with lower exposure had 42.9% fear, those with higher exposure significantly increased to 61.4% (*p*-value = 0.018).

Finally, the means of the factor analysis dimensions were compared for each group of secondary variables in Table 5. It was found that D3, mainly associated with job variables, had significant variations in age (*p*-value = 0.000) and employment relationship (*p*-value = 0.001), so people with more age and permanent contract showed better results in the dimension associated with the job.

## 4. Discussion

This study aimed to determine and describe the psychosocial health impact of the COVID-19 pandemic on ICU nurses and identify related risk factors. Our results show how the pandemic has generated thoughts in nurses associated with the lack of resources, knowledge, and means to alleviate the pandemic, as well as the feeling that the Spanish health system was not prepared for this situation. Despite this, many nurses had not lost their vocation and even considered their working relationship with colleagues to have been better, which speaks of resilience after the pandemic.

Regarding the aspects associated with the psychosocial dimension of mental health, our study shows that Spanish professionals felt much more worried, tired, and anxious than usual, in addition to having sleep problems. These results are confirmed by other studies [7], in which professionals had anxiety and depressive disorders, as well as emotional exhaustion. Numerous studies show a high prevalence of stress, anxiety, depression [13,15,18,23,24,25,26], sleep disorders [27] (such as insomnia [24,25,28]), exhaustion [13,28], and worry [19]. Another study [7] identified the shortage of PIDS as one of the causes of developing anxiety.

Fear was also a factor present in our study. There was an evident increase in fear in the nursing profession, both from the perspective of self-fear and the fear of infecting a family member. This aspect was also present in other studies [27,28] and identified as a more specific stressor [20,29], which adds a risk factor to the consequent appearance of a mental health problem.

Other causes that can be added to the occurrence of mental problems have much to do with the professional performance of an ICU nurse. A total of 92.8% of the participants felt that the pandemic affected their relationships, mainly due to the fear of infecting people in their social and family environment, as well as being subjected to isolation. According to another study, low levels of social support and rejection are recognised as risk factors for stress [13]. In addition, 94.1% had to take isolation measures because of their immediate environment, which several studies have identified as one of the triggers for mental distress [20], PTSD, depression, stress, and anxiety [13]. These measures of social isolation during the pandemic have been more severe for frontline professionals, not only from the perspective of the professionals themselves, but also from those known to be working in the ICU, which has led to assaults and contempt for health workers [5].

A key fact associated with psychosocial health is psychological support. A total of 18% of nurses have sought psychological or psychiatric help, and 37.1% have not, but think they need it. Several studies propose psychological support as one of the interventions to be followed to prevent the psychological impact of the pandemic on professionals [15,30,31]. From this perspective, it is necessary to increase Spanish nurses’ knowledge of the psychological support services in their hospitals, as almost half of them (47.6%) are not aware of these services.

In addition to these global data, some characteristics have been detected that are associated with these psychosocial health risks. Estos datos han sido referenciados en España por otros estudios, pero no asociados específicamente a enfermeras en ICU [32]. Being female is a factor associated with a worse impact of COVID-19 on mental health, as the women in our study have shown greater anxiety. This is supported in the literature by articles indicating that women are more likely to internalise trauma, resulting in a higher degree of psychological problems [33,34,35].

Being younger has a more significant impact on concentration at work, and the younger age subgroups were more likely to feel down and more anxious. On the other hand, they also felt that their work performance had been most negatively affected, and they the ones who were most likely to question their vocation. Recent studies have shown that age is related to a more significant impact of the pandemic on mental health. Thus, being younger is considered a significant risk factor for developing mental disorders, such as burnout, anxiety, PTSD, or psychological distress [34,35]. The cause of this more significant impact on younger professionals is related to less previous work experience [29,35], whereby inadequate staff training and little experience are also considered stressors.

Regarding the employment relationship, a contract with greater consolidation indicated less difficulty focusing. However, this group felt more tired, as well as feeling more depressed in a higher percentage. Different studies have linked experience and consolidation in employment with the development of coping strategies that reduce stress, emotional overload, and the ensuing impacts of such stressors [36,37]; so, despite feeling more depressed and fatigued, the permanent employees were able to maintain their concentration better.

Nursing professionals who had people at risk in their environment have also shown greater concern, and a higher percentage have required psychological help. Other studies support these results, supporting that those people who have children or close relatives with a higher risk of disease show greater concern about the situation [36,38].

In addition, those who had higher perceived exposure to COVID-19 have had higher levels of fear and psychological stress. According to the studies reviewed, the degree of exposure is the most critical risk factor for developing a mental disorder. It is associated with PTSD, disabling mental disorder, depression, anxiety, stress, sleep disorders, and burnout [5,9,13,19,20,27]. In addition, as we have seen, protective measures in our country have been deficient, causing an additional risk factor for those most exposed, especially in the case of ICU nurses.

The structure of the questionnaire showed a factor clearly associated with difficulties at work. It has been found that the better the contractual relationship and older the age, the fewer the difficulties associated with the job. As mentioned in other studies [36] in ICU nurses, experience is one of the determining factors in the management of situations. If good contractual conditions are added to this, a favourable context for the improvement of the work experience and, consequently, psychosocial health of nurses arises.

As has been seen, the present study provides interesting information on the variables associated with psychosocial health risk factors in nurses; however, it has some limitations. Due to the situation in which it was constructed (confinement and teleworking), no specific previous validation studies were carried out. However, the questionnaire was previously passed to several health professionals who reviewed its content and pointed out relevant modifications. More precise studies are needed to review in-depth the psychometric properties of the questionnaire and its possible suitability for similar pandemic situations in the future. Additionally, due to the need to obtain responses in a simple manner from the target population and considering the pandemic situation, the study was conducted online, which has some advantages associated with the rapid collection and management of data. However, the online questionnaire does not allow for a reliable sampling of the entire population of ICU nurses in Spain, making the results obtained descriptive; the same cannot be inferred for the rest of the Spanish ICU nurses.

## 5. Conclusions

The present study indicates that the COVID-19 pandemic has had a significant effect on Spanish ICU nurses, in terms of worry, fatigue, sleep problems, fear, anxiety, and stress. In particular, those nurses who were female, younger, and with a temporary employment contract were shown to be more vulnerable to the impact of the pandemic on their psychosocial health. In addition, those nurses who had people at risk in their environment and a greater degree of exposure to the virus in their workplace suffered a greater impact to their psychosocial health. In addition, a high percentage of professionals felt that their work performance had been negatively affected and experienced difficulties in concentrating, which impacted the quality of health care.

These results highlight the importance of exploring psychological distress among healthcare workers, as it may have long-term implications for personal and professional well-being. Our study sheds light, so that future research can build on it to design and adopt effective psychological impact prevention strategies and measures to protect nurses’ psychosocial health. The risk factors associated with psychological problems have been identified and should guide future interventions to improve nurses’ mental health.

Given some of the relationships identified, such as the importance of contractual stability, these intervention measures must be carried out, not only by the health system, but also the state, which is also involved in ensuring that in future similar situations, the protective measures and working conditions of frontline staff are the best possible. This will result in improvements in employment, the health of patients, and the psychosocial health of the workers.

## Figures and Tables

**Figure 1 healthcare-10-00796-f001:**
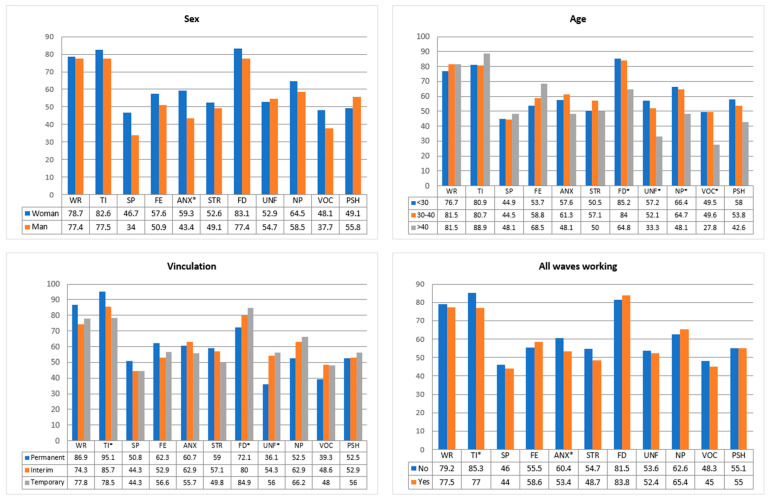
Association between psychological variables and nurses’ characteristics. WR (worried), TI (tired), SP (sleep problems), FE (fear), ANX (anxiety), STR (stress), FD (felt down), UNF (unfocused), NP (negative performance), VOC (vocation), PSH (psychological kelp); * significant value.

**Figure 2 healthcare-10-00796-f002:**
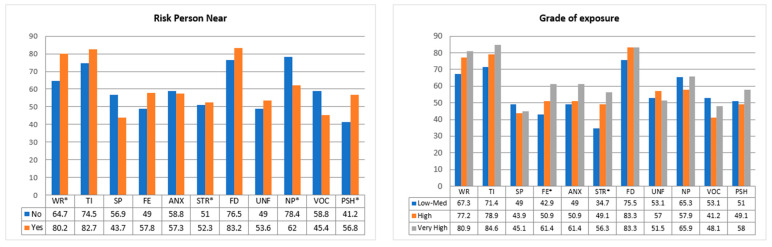
Association between psychological variables and nurses’ characteristics. WR (worried), TI (tired), SP (sleep problems), FE (fear), ANX (anxiety), STR (stress), FD (felt down), UNF (unfocused), NP (negative performance), VOC (vocation), PSH (psychological help); * significant value.

**Table 1 healthcare-10-00796-t001:** General characteristics of surveyed nurses (*n* = 456).

Variable	*n* (%)
**Sex**	
Woman	403 (88.4)
Man	53 (11.6)
**Age**	
<30 years	283 (62.1)
30–40 years	119 (26.1)
>40 years	54 (11.8)
**Employment relationship**	
Temporary	325 (71.3)
Interim	70 (15.4)
Fixed	61 (13.4)
**Indicate in which wave of the pandemic you have worked in the Intensive Care Unit**	
In all waves	191 (41.9)
Only in some	265 (58.1)
**COVID-19 infection**	
Yes	101 (22.1)
No	355 (77.9)
**Person at risk in close proximity**	
Yes	405 (88.8)
No	51 (11.2)
**What do you consider your level of exposure has been within the unit?**	
Very high	290 (63.6)
High	114 (25)
Medium	38 (8.3)
Low	11 (2.4)
Very Low	3 (0.7)
**How has the relationship with your colleagues been?**	
Better	352 (77.2)
The same as always	62 (13.6)
Worst	42 (9.2)
**Do you think you have been provided with adequate working conditions to cope with the situation since the beginning of the pandemic?**	
Yes	97 (21.3)
No	359 (78.7)
**Have you been provided with all the assistance you have needed, both material and personnel?**	
Yes	147 (32.2)
No	309 (67.8)
**Do you think our healthcare system was prepared for this situation?**	
Yes	7 (1.5)
No	449 (98.5)
**Have you experienced any traumatic events related to the pandemic?**	
Yes	169 (37.1)
No	287 (62.9)

**Table 2 healthcare-10-00796-t002:** Thoughts and feelings during the pandemic (*n* = 456).

Variable	*n* (%)
**Have you felt more worried (either about your health or your family’s health, work or anything related to the pandemic)?**	
Much more	358 (78.5)
More	89 (19.5)
Equal	8 (1.8)
No	1 (0.2)
**Have you experienced difficulties in focusing on your work?**	
Yes	242 (53.1)
No	214 (46.9)
**Have you felt more tired?**	
Much more	373 (81.8)
More	80 (17.5)
Equal	2 (0.4)
No	1 (0.2)
**Have you had sleep problems?**	
Much more	206 (45.2)
More	184 (40.4)
Equal	20 (4.4)
No	46 (10.1)
**Have you ever felt fear? To what degree?**	
Much more	259 (56.8)
More	178 (39)
Equal	12 (2.6)
No	7 (1.5)
**Have you experienced fear of infecting your close family/social environment?**	
Yes	445 (97.6)
No	11 (2.4)
**Do you feel that your relationships have been affected to some extent by the situation you have faced?**	
Yes	423 (92.8)
No	33 (7.2)
**Have you changed your behaviour towards those close to you, or even felt compelled to take isolation measures by them?**	
Yes	429 (94.1)
No	27 (5.9)
**Indicate the degree to which you have felt anxiety.**	
Much more	262 (57.5)
More	156 (34.2)
Equal	17 (3.7)
No	21 (4.6)
**Do you consider that you have suffered psychological stress and to what degree?**	
Much more	238 (52.2)
More	186 (40.8)
Equal	11 (4.6)
No	21 (2.4)
**Have you ever felt down, depressed or hopeless?**	
Yes	376 (82.5)
No	80 (17.5)
**Have you had times when you felt you had lost your vocation or even considered giving up your job?**	
Yes	214 (46.9)
No	242 (53.1)
**Do you think this has had a negative impact on your job performance, and therefore on the quality of patient care?**	
Yes	291 (63.8)
No	165 (36.2)
**Are you aware of the counselling services offered in your hospital to professionals?**	
Yes	239 (52.4)
No	217 (47.6)
**Have you had to seek psychological or psychiatric help?**	
Yes	82 (18)
No	205 (45)
No, but I think I need it	169 (37)
**Has the use of psychotropic medicines increased?**	
Yes	83 (18.2)
I consume the same	33 (7.2)
Non-consumption	340 (4.6)

**Table 3 healthcare-10-00796-t003:** Dimensions of the factor analysis of the primary variables.

Question	D1	D2	D3
Have you felt more worried (either about your health or your family’s health, work or anything related to the pandemic)?	0.808		
Have you felt more fear?	0.805		
Have you felt more tired or exhausted?		0.507	
Have you had sleep problems?		0.636	
Indicate the degree to which you have felt anxiety.		0.727	
Do you consider that you have suffered psychological stress, and to what degree?		0.783	
Have you ever felt down, depressed, or hopeless?		0.495	
Have you had to seek psychological or psychiatric help?		0.685	
Have you experienced difficulties in focusing on your work?			0.696
Do you think this has had a negative impact on your job performance, and therefore on the quality of patient care?			0.817
Have you had times when you felt you had lost your vocation or even considered giving up your job?			0.507

**Table 4 healthcare-10-00796-t004:** Results of chi-square tests comparing grouped and covariate factors.

	Sex		Age		Employment Relationship		All Waves Working		Risk Person Near		Degree of Exposure	
Question	Chi^2^	*p*-Value	Chi^2^	*p*-Value	Chi^2^	*p*-Value	Chi^2^	*p*-Value	Chi^2^	*p*-Value	Chi^2^	*p*-Value
Worried	0.047	0.828	1.481	0.477	3.361	0.186	0.203	0.652	6.484	0.011 *	4.718	0.095
Tired	1.612	0.204	2.072	0.355	10.381	0.006 *	5.160	0.023 *	2.049	0.152	5.752	0.056
Sleep problems	3.044	0.081	0.222	0.895	0.906	0.636	0.190	0.663	3.167	0.075	0.368	0.832
Fear	0.838	0.360	4.322	0.115	1.199	0.549	0.454	0.501	1.416	0.234	8.075	0.018 *
Anxiety	4.850	0.028 *	2.652	0.265	1.504	0.471	2.209	0.137	0.044	0.834	5.355	0.069
Stress	0.236	0.627	1.586	0.452	2.543	0.280	1.615	0.204	0.034	0.854	8.438	0.015 *
Felt down	1.077	0.299	13.251	0.001 *	6.155	0.046 *	0.392	0.531	1.422	0.233	1.831	0.400
Unfocused	0.065	0.798	10.470	0.005 *	8.244	0.016 *	0.067	0.795	0.378	0.539	0.990	0.610
Negative performance	0.736	0.391	6.620	0.037 *	4.204	0.122	0.378	0.539	5.312	0.021 *	2.314	0.314
Vocation	2.035	0.154	9.022	0.011 *	1.634	0.442	0.478	0.489	3.261	0.071	2.395	0.302
Psychological help	0.869	0.351	4.426	0.109	0.420	0.811	0.001	0.989	4.462	0.035 *	2.985	0.225

* Significant value.

**Table 5 healthcare-10-00796-t005:** Results of mean comparation tests between dimensions and covariate factors.

	Sex		Age		Employment Relationship		All Waves Working		Risk Person Near		Degree of Exposure	
Dimension	t	*p*-Value	F	*p*-Value	F	*p*-Value	t	*p*-Value	t	*p*-Value	F	*p*-Value
D1	0.448	0.654	2.715	0.067	1.679	0.188	0.306	0.759	1.867	0.062	2.582	0.077
D2	1.821	0.069	0.868	0.420	0.694	0.500	−1.450	0.148	0.568	0.570	2.396	0.092
D3	0.244	0.808	8.926	0.000 *	6.813	0.001 *	0.641	0.522	−1.116	0.265	0.285	0.752

* Significant value; t—statistic when two groups; F—with more than two groups.

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
