# Peer review of "Analysis of the Psychosocial Impact of the COVID-19 Pandemic on the Nursing Staff of the Intensive Care Units (ICU) in Spain"

_healthcare, 2022, doi:10.3390/healthcare10050796_

Round 1

Reviewer 1 Report

Comments to the Author:

This manuscript presents data on the psychosocial impact of the COVID-19 pandemic on the nursing staff of the ICU. In this unprecedented pandemic situation, the present study is of high importance. However, there are several issues that need to be addressed:

  1. Description of study participants are not clear. In 2.3 inclusion criteria for study sample is -Being healthcare personnel in an Intensive Care Unit; However, not all healthcare personnel in ICUs are nursing staff.
  2. Lines 170-171 – “…frequencies of 170 less than 5% were grouped to obtain groups with good subjects for statistical testing” – how such grouping affected results? for how many variables and how many subgroups were merged for each variable?
  3. Table 1 – questions – “Do you think they have been provided with adequate….” and “have they been provided with all the assistance…” – who are “they”?
  4. Tables - Table 2 on line 227 is cited wrongly; Table 3 is not cited at all.
  5. Table 3 – What “Linking” means in this table?
  6. Most questions on perception of the state of mental health had different answer options. How chi-square tests, presented in table 3, have been done?
  7. In Fig. 1 - what TE (Fear or Fear) means?
  8. In Fig.1 – Vinculation categories should be in English
  9. Fig 1 and Fig 2 – needs titles before description of abbreviations.
  10. Discussion – is more re-describing results; what the results show in light of the current literature? what is your main conclusion? what can be suggested?
  11. Study limitations should be desribed.
  12. Several English edits are required.
  13. Annex 1 is in Spanish.

Author Response

Thank you very much for your comments on the article. In view of your comments, the following changes have been made: 
1. The description of the participants has been changed.
2. The description of the statistical methods has been improved by expanding the information with the chi-squared test used and with the groupings made.
3. Suggested changes have been made to tables and figures and annex 1 has been translated into English.
4. The discussion has been expanded and improved, adding new references and including the limitations of the study.
5. The conclusions of the study have been included.
6. Possible language errors have been reviewed and corrected.

Reviewer 2 Report

Dear authors, thank you for this manuscript. However, it must be improved.

  1. English - the manuscript must undergo extensive English modifications in order to be suited for a professional journal. The manuscript is currently written in a very plain manner. Some parts are written in past tense, some in future tense...e.g. line 154: Participants who do not express their consent to participate in the study at the begin- 154
    ning of the questionnaire will be excluded
  2. Introduction: Line 80: In our research, we wanted to describe - please send to english editing services.
  3. Introduction: Line 78 - As we have seen, several studies have shown that the prevalence of stress, anxiety 78
    and depression among frontline healthcare workers is high [10, 11] - please consider additional health care professionals who dealt with elevated psychological distress at the same period of time - e.g. Shacham, M., Hamama-Raz, Y., Kolerman, R., Mijiritsky, O., Ben-Ezra, M., & Mijiritsky, E. (2020). COVID-19 Factors and Psychological Factors Associated with Elevated Psychological Distress among Dentists and Dental Hygienists in Israel. International journal of environmental research and public health17(8), 2900. https://doi.org/10.3390/ijerph17082900 Giorgi, G., Lecca, L. I., Alessio, F., Finstad, G. L., Bondanini, G., Lulli, L. G., Arcangeli, G., & Mucci, N. (2020). COVID-19-Related Mental Health Effects in the Workplace: A Narrative Review. International journal of environmental research and public health17(21), 7857. https://doi.org/10.3390/ijerph17217857
  4. M&M section: Should be better arranged. Feels a bit too chaotic. Perhaps consider about moving some of the study's design limitations to the limitations section/paragraph. In addition, what are UCA abbreviation stands for? Line 178. 
  5. The authors discuss psychological distress in their results, but I didn't find any relation to this in the M&M section? In addition, perhaps a better statistical analysis should be conducted such as structural analysis of the unvalidated questionnaire used in this study.
  6. The authors defintion of anxiety and other psychological criteria are poorly defined; questions like "have you felt more anxious" is very generalized and are very subjective; The authors should have used better surveys for psychological indices, e.g. GAD-7 and others.
  7. Limitations section/paragraph is missin.
  8. Conclusions section is missing. The authors say they conclude something, and then continue with reporting additional results. 
  9. No take-home message is given. It seems like the authors forgot about that matter. Reporting raw data solely is insufficeint for manuscript publication.
  10. References: The references should be better suited to MDPI's format...please review the author guideliens.

I would be glad to review a revised version of this manuscript. Hopefully, the authors will do a thorough work on it.

Author Response

Thank you very much for your comments on the article. In view of your comments, the following changes have been made:

1. Possible errors in the language have been checked and corrected.
2. New references have been added, including some of those mentioned by the reviewer.
3. The methods section has been reorganised. A comment on limitations has been added at the end of the article.
4. The description of the construction of the questionnaire has been improved. Statistical analyses on the properties of the questionnaire have been added.
5. The conclusion section has been added and the discussion section has been extended and improved with new references.
6. The format of the references has been changed to the MDPI format.

Reviewer 3 Report

First of all, congratulations for the work done, it is a research of great interest at the moment. After reviewing the manuscript, the following points could be improved:

The introduction should be improved, it is too brief and does not adequately address the interest of the study and its rationale. There are studies in our country on the subject that can be consulted (two recent ones are attached). It should be mentioned in the introduction and/or discussion.

https://doi.org/10.3390/ijerph18020580

https://doi.org/10.1111/inr.12748

Psychometric characteristics of the instrument used could be included.

Finally, I think it would be interesting to include in the manuscript a separate section on the "limitations of the study". 

Best regards.

Author Response

Thank you very much for your comments on the article. In view of your comments, the following changes have been made:

1. The introduction has been improved and expanded by incorporating some of the references mentioned by the reviewer.
2. Some psychometric properties of the instrument used have been included.
3. A paragraph has been added at the end of the discussion with the limitations of the study.

Reviewer 4 Report

The main drawbacks of the paper are the following:

The authors should consider rewriting the abstract section. The research ideas could have been more effective through the use of elaborative and concise sentences. The abstract, as is, does not provide a concise account of the work and conclusion of the research study. It needs to be more structured and synthesized for research clarity. It is necessary to provide a more straightforward explanation of how research data was obtained. It is essential to mention the instruments within the abstract with a quick overview of the number of areas covered by the research design, instrument, data, and methodology in detail.

The literature review or Background section is missing. This section has the role of presenting other theories related to the topic and explaining in further detail the research gaps that the paper seeks to close and why the paper is needed to recognize the current gaps in the literature.

Section 2 Materials and Methods is the main weakness of the article. We did not find any research questions or hypotheses.
The results were not well-presented to readers to understand the focus of the research study.

The results must be interpretive rather than just descriptive and connect the research results with relevant literature citations for validity and reliability.

The Discussion is not well-presented as it does not integrate with the research study results to provide a coherent scholarly argument.

The Conclusion section is missing.

The subject is interesting, but the sources are limited to only 23 references.

Good luck!

Author Response

Thank you very much for your comments on the article. In view of your comments, the following changes have been made:

1. The abstract has been rewritten following some of the reviewer's recommendations.
2. The introduction has been expanded, adding new references and extending the content.
3. The results section has been changed and results relating to the psychometric properties of the questionnaire have been added.
4. The discussion has been extended with new references and the limitations of the study have been added at the end.
5. The conclusions section has been created.

Round 2

Reviewer 2 Report

I thank the authors for their kind review and corrections.

Author Response

Thank you very much for your reviews of the article. Regarding the issue of minor language revisions, several corrections have been made.

Specifically, in the results, the error that interchanged man with woman in table 1, page 5, has been corrected. The translation mistake in lines 111 to 114, page 2, has also been corrected. Minor language corrections were also made to the language of the lines 102, 105, 123, 181, 227, 281, 298, 305, 306, 311, 355, 358, 380, 405 and 408.

Reviewer 4 Report

Good luck!

Author Response

Thank you very much for your reviews of the article. The corrections suggested by the reviewers and the academic editor have been made with the aim of improving the sections that, according to this review, needed some improvement.

Regarding the issue of minor language revisions, several corrections have been made. Specifically, in the results, the error that interchanged man with woman in table 1, page 5, has been corrected. The translation mistake in lines 111 to 114, page 2, has also been corrected. Minor language corrections were also made to the language of the lines 102, 105, 123, 181, 227, 281, 298, 305, 306, 311, 355, 358, 380, 405 and 408.